# Learning Structure-Aware Representations of Dependent Types

**Konstantinos Kogkalidis**[1,2]
kokos.kogkalidis@aalto.fi

**Orestis Melkonian**[3]
orestis.melkonian@iohk.io

**Jean-Philippe Bernardy**[4,5]
jean-philippe.bernardy@gu.se

[1]Aalto University
[2]University of Bologna
[3]Input Output (IOG)
[4]University of Gothenburg
[5]Chalmers University of Technology

## Abstract

Agda is a dependently-typed programming language and a proof assistant, pivotal in proof formalization and programming language theory. This paper extends the Agda ecosystem into machine learning territory, and, *vice versa*, makes Agda-related resources available to machine learning practitioners. We introduce and release a novel dataset of Agda program-proofs that is elaborate and extensive enough to support various machine learning applications – the first of its kind. Leveraging the dataset's ultra-high resolution, which details proof states at the sub-type level, we propose a novel neural architecture targeted at faithfully representing dependently-typed programs on the basis of structural rather than nominal principles. We instantiate and evaluate our architecture in a premise selection setup, where it achieves promising initial results, surpassing strong baselines.

## 1 Introduction

Automation on theorem proving over arbitrary domains has been a staple task for computer science and artificial intelligence, and a concept that in fact predates the inception of both fields. Over the last decade, the success of gradient-based optimization and large-scale neural models has brought this once elusive goal within reach. Capitalizing on that fact, community efforts have led to a wider availability of language-specific theorem-proving datasets. This, in turn, has led to a constant stream of milestones and achievements. Rather than threatening theorists worldwide with premature retirement, this development empowers mathematicians and computer scientists on the individual scale, and potentiates an unprecedented acceleration for the formal sciences on the collective level.

Theorem-proving tasks can take on many forms, one of which is *premise selection*. In a premise selection setup, the model is presented with a goal statement and a collection of theorems or lemmas, some of which are potentially relevant for the proof process, whereas others are not. The model is then tasked with selecting which of those lemmas are relevant. A successful model can relieve burden from the human operator by dramatically reducing or even trivializing the proof search space. From an epistemic perspective, premise selection epitomizes the good aspects of neurosymbolic AI. Human and machine get to interact and collaborate towards a common goal. Meanwhile, the model's predictions are made safe by virtue of being confined within (and verified by) a trustworthy *white box*: the proof assistant.

Depending on the logic employed by the prover, the line between proof search and program synthesis may become blurry. Concretely, the Curry-Howard correspondence asserts an equivalence between intuitionistic logics and constructive type theories; proofs in such logics are indistinguishable from programs in some corresponding functional programming language, and *vice versa*. This marvel has been the foundational core behind the development of type-theoretic proof assistants, such as

38th Conference on Neural Information Processing Systems (NeurIPS 2024).

Coq, Lean and Agda, *inter alia*. On the one hand, these are programming languages, except strict enough to find use in the formalization of science or the certification of engineering. On the other hand, these are also theorem provers, except with a less obscure syntax, that offer additional niceties such as compilation-level reporting, termination checking, and a natural ability to extract executable programs from the proofs produced.

In the machine learning world, domain-, language- and task-specific representation learning for theorem proving has gained significant community attention in recent years. Modeling strategies can be broadly grouped in two general categories. The dominant approach relies on sequence-based encoder and/or decoder architectures, operating on the user-facing representation of formal objects (theorems, formulas, lemmas, *etc.*); this approach has become popular due to its direct compatibility with generic pretrained large language models. The alternative approach instead seeks to explicate the relations within and between these formal objects, *e.g.,* with tree or graph architectures.

## 1.1 Contributions

In this paper, we identify and address two weak points in the existing literature.

**Machine Learning for Agda** First, we note a disparity in the availability of machine learning resources and results across proof assistants. A plurality of datasets, models and tools are available for a handful of languages, the majority of them for Coq, HOL and Lean, whereas none exist for Agda. We believe that this state of affairs is an effect of historical momentum and methodological precedence, rather than inherent merit. Agda spearheads developments in constructive type theory with an array of innovative features, gaining adoption and a rapidly maturing ecosystem. Aiming to aid Agda with meeting and expanding her potential, we implement an Agda program that extracts intermediate compilation steps, and produces them in the form of human-readable partial proof states. Applying the extraction algorithm, AGDA2TRAIN, on established high-quality libraries, we obtain and release an elaborate and extensive dataset of Agda program-proofs, at a scale and form that can support various machine learning applications – the first of its kind.

**Modeling Type Structure** Second, we note a significant gap between the rigor of the structures modeled and the structural lenience of the architectures employed. Seeking to bridge this gap, we design and implement a general-purpose scheme for the faithful representation of expressions involving dependent types. We apply our methodology on the extracted dataset to produce QUILL: a novel neural guidance tool for Agda. Beyond its current instantiation in Agda, the system is *universal*, being equally applicable to any language that uses type theory as its foundational core.

## 2 Background

### 2.1 Proof Assistants

The leading proof assistants (Coq, Lean, Agda) all follow the same core principles with respect to the kinds of objects they define and manipulate. Modulo theoretical flavorings and syntactic sweeteners, these are always the terms of a typed $\lambda$-calculus. Leveraging the *propositions-as-types* interpretation [Sørensen and Urzyczyn, 2006, Wadler, 2015], the proof assistant's duty is chiefly to check that a program adheres to a specific *type*, or, equivalently, that a proof attests to a particular *proposition*. Typically, proofs are constructed in an incremental fashion: the user can defer some sub-proof to the future by instead supplying a *hole*. We provide an illustrative example in Appendix A, Figure 4. A proof is only complete if it is without holes. The proof assistant may assist the user by providing a set of hypotheses (or premises), which can be used to fill in a given hole. The problem of singling out relevant premises is completely specified by (i) the type of the hole itself (henceforth called the *goal*), and (ii) the types of premises available. Except for trivial problems, the set of premises that can be used in any given hole is enormous, and the number of ways in which they can be combined is even larger. Being able to filter and rank premises thus goes a long way towards efficient proof search. To aid in the process of proof construction, some assistants additionally implement *tactics*. Tactics can range from simple (introducing a single $\lambda-$abstraction) to complex (type-guided combinatorial search, domain specific solvers, *etc.*). Tactics also get to heavily benefit from a premise selection pass, given their combinatorial nature; however, they do not (yet) play a major role in the Agda world, and are not of primary interest to us here.

## 2.2 Related Work

Navigating the search space of possible proofs by purely symbolic means is generally infeasible. As an alternative, several lines of work have explored the application of statistical learning methods as a way to either truncate the search space, or to guide the theorem prover.

An early step of theorem proving into machine learning territory is HolStep [Kaliszyk et al., 2016]. HolStep provides string representations of intermediate proof states extracted from 11,400 narrow-domain but non-trivial HOL proofs. The dataset is used to benchmark character- and token-level encoders in the task of telling whether a statement is relevant in the proof of a conjecture or not. Another contribution along the same lines is HOList [Bansal et al., 2019]. The dataset encompasses 30,000 HOL proofs, and the task is framed as selecting a tactic and its arguments, given a goal. The modeling approach rests on DeepHOL: a reinforcement learning loop incorporating convolutional encoders applied on string-formatted formulas.

On the type-theoretic front, two related and concurrent contributions are GamePad [Huang et al., 2018] and CoqGym [Yang and Deng, 2019]. Both implement a minimal interactive interface with Coq, the latter also exposing a dataset of 71,000 human-written proofs. The main task is once more framed as tactic selection. GamePad uses a sequential recurrence to encode formula terms, emulating variable binding using a random embedding-association table. CoqGym, on the other hand, employs a tree-recursive encoder for representing goals and premises, and a semantically constrained tree-recursive decoder for decoding into the tactic language.

Beginning with the early works of Urban and Jakubův [2020] on Mizar and Polu and Sutskever [2020] on Metamath, large language models (LLMs) have by now permeated the theorem proving literature. Interfaces between LLMs include LeanStep [Han et al., 2021], lean-gym [Polu et al., 2023] and LeanDojo [Yang et al., 2023] for Lean, as well as Lisa [Jiang et al., 2021] and Magnushammer [Mikuła et al., 2023] for Isabelle. LeanStep and lean-gym export and utilize compiler-level type information from the intermediate proof steps of about 128,000 proofs; yet these are still formatted as strings and processed as such by the corresponding LLM. LeanDojo exports similarly-structured data from 98,734 theorems, additionally providing interactive capabilities, while Lisa exports shallow states from 183,000 theorems, represented as sequential MDPs. Magnushammer, finally, provides access to a shallow premise dataset from 570,000 mixed human and synthetic proofs states. The models underlying the above works all build on the same key idea: an LLM is exposed to a textual representation of an incomplete proof, and is then tasked with autoregressively continuing.

Evidently, the bulk of the modeling work has so far been outsourced to sequential architectures. Despite their merits, we argue in Section 4.2 that such models are *ill-equipped* to capture the structures being modeled, except superficially and on a data-driven basis. The structural effects of (co-)reference, variable binding and abstraction, equivalence under substitution *etc.* are all dismissed in favor of a uniform and simplified representation format. Other than aforementioned exceptions, works that stand out in this respect are those of Wang et al. [2017] and Paliwal et al. [2020], who opt for a message-passing approach on formula trees (using graph-like edges to handle co-referencing and variable binding). Along the same lines, Li et al. [2020] use a 2-level hierarchical encoder to distinguish between intra- and inter-formula attention. Contemporary work [Blaauwbroek et al., 2024] further explores this perspective, fixing a symbol embedding table for a collection of symbols and employing message-passing networks over formula graphs to encode definitions in Coq. Representations are built in topological order, such that referring objects can utilize the previously computed representations of their referents. These representations finally find use in allowing the dynamic prediction of tactic arguments, given a chosen tactic.

Our contribution follows along and expands upon the structurally-disciplined line of approaches. Our data-generation process explicates the structure of Agda files at the finest possible resolution: that of the internal type structure. Like Blaauwbroek et al. [2024], our modeling approach iteratively builds dynamic object representations that are derived on the basis of structural rather than nominal principles. Unlike Blaauwbroek et al. [2024], our representations are also: (1) *complete*, in the sense of allowing the representation of any item in the object language, and (2) respectful of $\alpha$-equivalence, in the sense of being *invariant to variable renaming and substitution*.

## 3 Data

Data is extracted from Agda files in a type-mindful manner. We first allow Agda to type-check program-proofs as usual, and then consult her for their internal representations, which we store mostly

```
open import Relation.Binary.PropositionalEquality using (_≡_; refl; cong; trans)

data ℕ : Set where          _+_ : ℕ → ℕ → ℕ
  zero : ℕ                  zero  + n = n
  suc  : ℕ → ℕ              suc m + n = suc (m + n)

+-comm : (m n : ℕ) → m + n ≡ n + m
+-comm zero     zero    = refl
+-comm zero     (suc n) = cong suc (+-comm zero n)
+-comm (suc m) zero     = cong suc (+-comm m zero)
+-comm (suc m) (suc n) = cong suc (trans (+-suc m n) (+-comm (suc m) n))
    where +-suc : ∀ m n → m + suc n ≡ suc (m + n)
          +-suc zero    n = refl
          +-suc (suc m) n = cong suc (+-suc m n)
```

Figure 1: Agda code formalizing the commutativity of addition.

unchanged. These representations are formally safe and unambiguous, but also more honest than the surface pretty prints we get to witness as users, as syntactic sugar has been translated away.

Figure 1 presents our running example: a mechanised proof that addition on natural numbers is commutative. The program first imports the standard library's propositional equality ($\equiv$) and some of its properties (refl, cong, trans). It then goes on to define Peano natural numbers as an inductive datatype (ℕ) and arithmetic addition as a recursive function (+) via pattern matching: a natural is either a 0 (zero) for which $0 + n = n$, or the successor (suc) of a number $m$, for which $(m + 1) + n = (m + n) + 1$. With the above in hand, we can inductively prove that addition is commutative (+-comm), *i.e.,* $\forall\, m\, n.\, m + n = n + m$, following a case-by-case analysis:

1. If $m$ and $n$ are both 0, we can prove our goal using reflexivity: both sides of the equation are syntactically equal.

2. If $m$ is 0 and $n$ is the successor of some other $n'$, then we can invoke the proof of the commutativity of $m + n'$.

3. Similar to (2): if $n$ is 0 and $m$ is the successor of some other $m'$, then we can invoke the proof of the commutativity of $m' + n$.

4. If both $m$ and $n$ are successors of $m'$ and $n'$, respectively, then we appeal to a helper lemma about how the successor function distributes across addition (+-suc), before finally invoking the proof of the commutativity of $m + n'$.

Note how each case above corresponds to a pattern-matching clause in the function definition and inductive hypotheses manifest as recursive function calls; that is the beauty of dependently-typed functional programming through the lens of the Curry-Howard correspondence.

## 3.1 Implementation

We implement a generic dataset extractor for arbitrary Agda programs as a *compilation backend*, AGDA2TRAIN, that now "compiles" a source Agda file to a corresponding JSON structure[1]. A collection of such structures together make up our extracted dataset.

## 3.2 Extracted Structure

Each extracted JSON file mirrors Agda's internal compilation state for a given file. This state is however neither static nor singular; it changes depending on what the current focal point of the compilation is. From the user's perspective, each state corresponds to a different moment in the coding process, where a subset of the file has been written, another exists only as a hole, and another is yet to be written altogether. We navigate through *all* such possible states. Concretely, we iterate through each of the definitions in a file, pausing *not* at the possible prefixes of a proof (seen as a textual object), but rather at all its possible sub-terms (seen as a syntactic tree). This allows us to optimally

---

[1] Capitalizing on Agda's extensible backend infrastructure, we develop AGDA2TRAIN as a standalone Haskell package, independent of the core Agda compiler. The package is publicly accessible at https://github.com/omelkonian/agda2train.

maximize the utilization of each file as a data resource, akin to a preemptive data augmentation routine (except type-safe, courtesy of Agda). At each step, we record the missing term that fills the hole, as well as the full *typing context* available at the time.

**Typing Context**   The typing context of a hole consists of its type (the goal), the collection of antecedent definitions in *scope*, plus a context of locally bound variables. For each definition and variable, we record its name, its type as well as its proof term.

**Term Structure**   Terms, whether type- or proof-level, are structurally explicated to the fullest by the JSON structure. Since Agda is a dependently typed language, its terms go far beyond typical $\lambda$-expressions that canonically consist of just abstractions, applications and variables. Terms capture Agda's internal grammar in its entirety, allowing the specification of many more constructs, such as datatype declarations, records, function clauses defined via pattern matching, patterns *etc.*. For the sake of readability (and to appease practitioners more interested in the sequential format), the dataset also includes pretty-printed displays of each AST.

For additional details on the extraction, including an example JSON matching the program of Figure 1, refer to Appendix A. For the reference JSON grammar, refer to Appendix C.

## 4   Modeling

### 4.1   Preliminaries

We begin by recalling that a snapshot of an interactive Agda session can be thought of as an incomplete but type-checked file. The file consists of a collection of definitions (henceforth lemmas), *i.e.,* all proofs or programs within scope (either imported or locally defined). Of these, zero or more might contain holes, *i.e.,* remain unfinished. The distinction between lemmas and holes essentially boils down to the presence or absence of a proof-level term. For the time being, we ignore those, and focus solely on type declarations. Lemmas and holes are then syntactically indistinguishable, since their types adhere to the same grammar.[2] Capitalizing on this symmetry, we represent both kinds of objects using a single parameter-sharing encoder. Given a collection of ASTs as input (some being lemmas, others holes), the encoder will be tasked with returning a vectorial representation for each.

### 4.2   Desiderata

**Conceptual**   Our primary aim is generalizability, which entails first *completeness* (*i.e.,* ability to model any data point in the domain, regardless of distributional shifts), and then appropriate *inductive biases* (*i.e.,* ability to effectively extrapolate from the patterns in the training data). The commonly employed sequential approach attains completeness by tokenizing expressions according to the co-occurrence frequencies of their constituent substrings, either at the "word" or at the "subword" level. In doing so, however, it provides the model with the *wrong* inductive biases, since the expressions under scrutiny are not formed by just *any* string grammar, but by precisely constrained inductions. At first glance, one might think to resolve the problem by simply applying a tree- or graph-based architecture on Agda ASTs. The challenge lies in the fact that Agda types, being dependent, are considerably expressive. While simple types are composed of a fixed set of operators and propositional constants (base types), dependent types may contain quantifications (introducing variable bindings), and refer both to other lemmas in scope, and to any variables previously introduced. Case in point, all AST-encoding approaches to date assume a fixed symbol vocabulary. By treating (even just) the primitive nodes denoting constants and variables as strings, they implicitly concede that all occurrences of the same string carry the same denotational meaning. Inadvertently, this also implies that the substitution of such strings with others alters the final representations of the objects these strings occur within, even if such substitutions are *semantically void*. Worse yet, it means that completeness is sacrificed. Shifts in naming conventions, use of libraries that are stylistically distant or targeted at novel domains, *etc.*, might necessitate the evocation of symbols not present in the trained embedding tables, collapsing representations into meaningless generics (*e.g.,* multiple different entries might be made the same due to an excess of [UNK] tokens). A non-nominal resolution of these complex referencing patterns necessitates an extension of the traditional AST-encoding

---

[2]To be precise, a hole is natively specified as a goal type and a context, *i.e.,* a "telescope" structure of local variable bindings. We use the proof-theoretic equivalence between hypothetical reasoning and dependent functions to merge context and goal into a single type by folding the assumptions provided in the context to a nesting of dependent functions. Put simply, if the context is $x : A$ and the goal is $B$, the folded type of the hole is $\prod_{x:A} B$. This effectively homogenizes the representation format, but at the cost of elongating goal types.

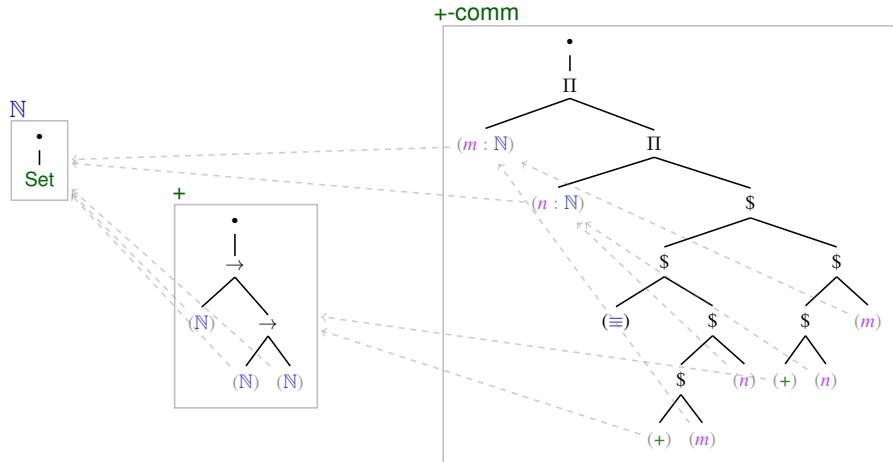

Figure 2: Tokenized view of lemma types from Figure 1; see Appendix B.1 for details.

approaches. Crucially, we need our encoder to generically refer to both locally defined variables (intra-AST) as well as other scope entries in their entirety (inter-AST); see Figure 2 for a visual example.

**Practical**  Beyond theoretical concerns, tree- and graph-based approaches are also lacking in several practical aspects. Tree-like approaches involve a bottom-up reduction of the AST into a single representation, often times achieved by translating leaves to vectors and operators to neural functions. Despite offering suitable biases, they struggle to capture long-distance relations, exhibit a temporal complexity that scales linearly with tree depth, and are challenging to parallelize across and within ASTs. In comparison, graph-based models are easier to parallelize, but also offer an appealingly native solution for non-tree edges. Moreover, by fixing the number of message-passing rounds, graph architectures become invariant to AST depth, thresholding temporal complexity. This, however, comes at a cost, as the model's perceptive field becomes significantly limited. Updated node features are obtained from a fixed-size local neighborhood, and their aggregation has no means to account for fine structure at larger scales, leading to a coarse and "lossy" (or oversmoothed) representation of the AST as a whole. As an alternative, Transformer-based encoders implementing full self-attention allow concurrent pairwise interactions between all nodes, regardless of their relative placements. This is particularly advantageous, as it equips the model with the means of capturing patterns and dependencies at arbitrary scales. But while this is true for sequences, a vanilla Transformer lacks the representational biases to perceive the complex inductive structure of an Agda file. Furthermore, its memory footprint scales linearly with the batch size, and quadratically with the longest sample's length. In a premise selection setup, each premise constitutes an "independent" sample, with the total of ASTs being processed making up a (subset of a) batch. This mandates operating on a file-by-file basis, which translates into a requirement for variable-sized batches, during both training and inference. Worse yet, AST counts and sizes tend to have a high variance, and can often be overlong, *e.g.,* when dealing with complex files or types. In brief, a vanilla Transformer is practically unusable; we will need to do better.

### 4.3  Architecture

Having identified a fully self-attentive encoder as an attractive modeling option, we set to amend the shortcomings of the Transformer with a modified architecture that is better attuned to the task. In the following paragraphs, we isolate and explicate its building blocks.

**Efficient Attention**  To first address the scaling issue, we adopt the linearized attention mechanism of Katharopoulos et al. [2020]. By foregoing the softmax normalization around the query-key dot-product matrix, this variant allows a sub-quadratic formulation of the attention function. Concretely, given an AST with $n$ input nodes, transformed into queries $q : \mathbb{R}^{n \times d}$, keys $k : \mathbb{R}^{n \times d}$ and values

$v : \mathbb{R}^{n \times e}$, we obtain an attention output $a : \mathbb{R}^{n \times e}$ given by:

$$a_i = \frac{\phi(R_i q_i) \cdot \left( \sum_{j=1}^n \phi(k_j R_j) \otimes v_j \right)}{\phi(R_i q_i) \cdot \sum_{j=1}^n \phi(k_j R_j)} \tag{1}$$

In the above equation, $\cdot$ and $\otimes$ denote the inner and outer product, applied here over dimensions $d$ and $(d, e)$ respectively. Following the empirical insights of Arora et al. [2023], we define the feature map $\phi : \mathbb{R}^d \to \mathbb{R}^{1+d+d^2}$ as:

$$\phi := x \mapsto [1] \; ; \; x \; ; \; \sqrt{0.5} \cdot \text{vec}(x \otimes x) \tag{2}$$

where ; denotes vector concatenation and vec flattens the $(d \times d)$-sized outer product into a $d^2$-sized vector. This formulation emulates a softmax-like exponentiation of the vector dot-product via a second degree Taylor polynomial. Finally, $R$ is a tensor packaging $n$ matrices from the orthogonal group $O(d)$ that serve to introduce a positional specification– we spell out their functionality in the next paragraph. In practice, the above operations are broadcasted over multiple ASTs $t$ and attention heads $h$, yielding an asymptotic complexity of $\mathcal{O}\left(tnehd^2\right)$. By setting $d \ll N$, where $N$ is the maximum AST size encountered, complexity remains tractable throughout training and inference.

**Structured Attention**    To enhance the model with the tree-inductive biases necessary to process ASTs, we employ the binary-tree positional encoding scheme of Kogkalidis et al. [2024]. The scheme alters the attention coefficients by rotating and reflecting query and key vectors with position-dependent orthogonal matrices. One such matrix is assembled for every unique AST position using a chain of matrix products made of two primitives. Each primitive, in turn, corresponds to choice of a tree branch option (*i.e.,* left or right). By applying the appropriate orthogonal matrix on each query $q_i$ and key $k_j$ prior to Taylor expansion, we practically modulate the dot-product between by the norm-preserving bilinear scalar function $R_i^\top R_j$ that represents the *relative* path between them upon the tree structure. This setup inductively parameterizes the construction of relative paths, allowing the representation of ASTs of arbitrary sizes and shapes using just two primitives.

**Static Embeddings**    Initially, we embed AST nodes depending on their content. Primitive symbols with a static, uniform meaning are handled by a simple embedding table. The table contains a distinct entry for each of four type operators (dependent function $\Pi$, simple function $\to$, $\lambda$-abstraction and function application), three terminal nodes, the content of which we designate as "don't-care" (sorts, levels and literals), and two meta-symbols (the start-of-sequence token [SOS], and an out-of-scope symbol [OOS], used to denote masked or otherwise unavailable references).

**Variable Representations**    Dependent function types and $\lambda$-abstractions introduce and bind variables to arbitrary names, which can then be referred to by nodes further down within the same AST. A canonical way to make the representations name-agnostic is de Bruijn indexing [de Bruijn, 1972], where names are replaced with an integer indicating the relative binder-distance to the variable's introduction. For representation learning purposes, substituting variable names with de Bruijn indices is not much good in itself. Despite the normalized vocabulary, there are still enumerably infinite indices to learn embeddings for. Furthermore, a single index carries no "meaning" of its own, other than as an address-referencing instruction. Even so, its meaning is highly contextual and indirect: the translation from binder-distance to a concrete "address" within the AST depends on the number and placement of variable binders within the referring node's depth-first ancestry. In order to facilitate learning, we take the extra step of actually *resolving* the indexing instruction, substituting the de Bruijn index with a pointer to the node introducing the referent. To represent the pointer, we then use the positional encoding scheme described earlier. We procure and compose the two orthogonal matrices representing the variable's and the binder's absolute positions to represent their relation, and apply the resulting matrix against a trainable embedding. The final vector enacts a representation of the variable's offset relative to its binder. This constitutes a name-free indexing scheme with a clean and transparent interpretation, abolishing the need for an ad-hoc nominal embedding strategy.

**Reference Representations**    Topologically sorted, the lemmas within a file (and, by proxy, their ASTs) form a *poset*. For lemmas $a$ and $b$, $a \leq b$ denotes that the transitive closure of lemmas referenced by $b$ contains $a$. Based on this ordering, the collection of ASTs can be partitioned into a strictly ordered collection of mutually exclusive subsets, each subset containing the ASTs of lemmas of a specific *dependency level*. We follow along this sequential order, using the encoder to iteratively

populate a dynamic embedding table of AST representations, using each AST's [SOS] token as an aggregate summary. Any time we encounter a node referencing another lemma, we can query the embedding table for the corresponding AST representation. Parallelism is still fully maintained within each partition, where entries do not depend on one another. From a high-level perspective, this makes for an adaptive-depth encoding strategy, implemented by a fixed-depth network.

**Putting Things Together**   Following standard practices, we compose an attention block with a SwiGLU [Dauphin et al., 2017, Shazeer, 2020], inserting residual connections and prenormalizing with RMSNorm [Zhang and Sennrich, 2019] to obtain a single layer. We compose multiple such layers together, sharing and reusing positional encodings across layers and attention heads.[3]

## 5   Experiments & Results

To quantitatively assess the model's capacity for representation learning, we instantiate it composed with a linear layer on top, and train it in a premise selection setting derived by minimally processing the dataset of Section 3. We call the composite model QUILL, reflecting on it being a featherweight tool (*n.b.*, Agda is a hen) intended to assist in the proof writing process[4]. QUILL takes a collection of ASTs as input, but produces now a scalar value for each hole-lemma pair, indicating the latter's relevance to the former.

### 5.1   Training & Evaluation

**Setup**   We extensively detail our exact experimental setup in Appendix B. Concisely, we train using InfoNCE on a size-constrained subset of the Agda standard library. Following some necessary processing and filtering steps, this amounts to 481 training and 92 evaluation files, counting 15,244 and 4,120 holes respectively. Out of the 92 evaluation files, 5 files are marked outliers with respect to the distribution of file sizes, containing 25% of the total holes and significantly exceeding the size of largest training file– we take them apart and treat them as a separate OOD evaluation set. To obtain more substantiated insights on OOD performance, we further consider two additional evaluation sets, extracted from portions of the Unimath [Rijke et al.] and TypeTopology [Escardó and contributors] Agda libraries. These are radically distant to the standard library in terms of content, abstraction level, proof style, library structure and file size. Both libraries, and especially so TypeTopology, are infamous for their austere and minimal (affectionately so-called spartan) style, making them ideal test sets at the furthest ends of the modeling domain. The portions we consider are extracts of 137 and 28 files with 5,247 and 1,983 holes for Unimath and TypeTopology respectively.

| | stdlib:ID | | stdlib:OOD | | Unimath | | TypeTopology | |
|---|---|---|---|---|---|---|---|---|
| MODEL | AVEP | R-PREC | AVEP | R-PREC | AVEP | R-PREC | AVEP | R-PREC |
| QUILL | **50.2**±0.5 | **40.3**±0.4 | **38.7**±0.7 | **31.1**±0.9 | **27.0**±0.4 | **17.4**±0.3 | **22.5**±0.3 | **15.4**±0.5 |
| no Taylor expansion | 47.0±0.4 | 36.2±0.5 | 37.1±0.5 | 29.2±0.6 | 26.8±0.2 | 17.0±0.1 | 21.4±0.5 | 14.4±0.4 |
| no Tree PE | 44.5±1.5 | 34.1±1.8 | 30.7±0.4 | 24.0±0.4 | 24.8±3.4 | 15.5±2.9 | 18.8±1.1 | 12.3±0.9 |
| no variable resolution | 35.8±2.7 | 25.9±2.7 | 25.5±2.7 | 19.1±2.4 | 19.7±1.8 | 11.6±1.5 | 17.7±3.0 | 11.0±3.1 |
| Transformer | 10.9±0.4 | 3.7±0.2 | 8.5±0.2 | 4.5±0.1 | 9.4±0.3 | 3.9±0.1 | 5.8±0.0 | 0.9±0.0 |

Table 1: Model performance under different ablations across all evaluation sets.

### 5.2   Results

To evaluate performance, we gather the pairwise matching scores for all hole-lemma pairs of the stdlib:ID evaluation set, group them by ground truth label and visualize them in Figure 3. On the standardized axis, we see negative pairs centered around the sample mean at 0. Even though there is overlap between the scores of positive and negative lemmas, 80% of the positive pairs are ranked significantly higher than 80% of negative pairs, with a wide separation between the lowest positive and highest negative quartile. What this means in practice is that the trained model enacts a relatively

---

[3]The orthogonal matrices are applied per-head, and are thus lower-dimensional than the encoder as a whole. This makes them superficially incompatible with the input embeddings. We fix the issue by lowering the embedding dimensionality, and expanding it later via a trainable linear map.

[4]Source code, configuration files and scripts supporting experiment execution are available at https://github.com/konstantinosKokos/quill.

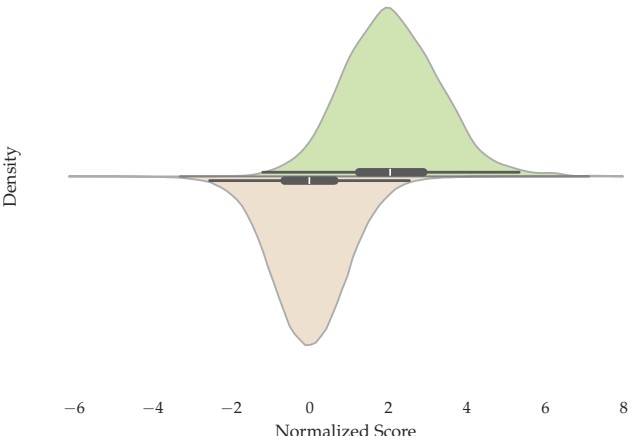

Figure 3: Empirical distributions of the standardized scores of relevant and irrelevant lemmas from the `stdlib:ID` evaluation set.

reliable discriminator, capable of discarding "strongly negative" lemmas (or, *vice versa*, filtering for "potentially positive" ones).

For a more quantitative evaluation, we compare the ranking of lemmas produced by QUILL with the set of relevant items defined by the gold standard labels. We borrow metrics from recommender system literature: average precision (AVEP) and R-precision (R-PREC). Intuitively, AVEP provides an estimate for precision as one iterates through suggestions until all relevant lemmas have been retrieved. R-PREC, on the other hand, assesses the model's effectiveness in placing the set of relevant lemmas (of any size $R$) at the top $R$ positions among the ranked candidates. We report results in Table 1 (means and 95% confidence intervals of 4 training repetitions).

Within the in-distribution evaluation set, we see relevant lemmas interspersed among as many irrelevant ones at the top of the ranked suggestions. This is particularly compelling when compared to the average ratio of relevant lemmas to total candidates, which lies below 1.5%. Unsurprisingly, performance degrades the further away one drifts from the original training distribution. The `stdlib:OOD` set contains ASTs and scopes that are approximately two orders of magnitude larger than the training mean. On top of ASTs being harder to informatively represent, the average ratio of relevant lemmas to available options is now about 2‰. The other two libraries essentially constitute zero-shot transfer learning setups, intended to stress test the model's representations.

**Ablations**   To experimentally validate our design decisions, we conduct an ablation study where we substitute network components for less exotic alternatives. We consider three ablations, first in isolation (*i.e.,* keeping the rest of the model intact), and then in combination. First, we completely skip the Taylor expansion step, opting for an offset-by-one ELU feature map instead. We note a modest but tangible performance hit, but drastically improved memory complexity. Second, we remove the tree-structure bias in the attention function, substituting the multiplicative tree-positional encoding with the additive sequential sinusoids of the vanilla Transformer. The performance effect is notable, as the model is burdened with the added cognitive load of internally "parsing" the sequentialized type expressions. Finally, we remove the structural de Bruijn resolution, treating all varables as syntactically equivalent and mapping them to the same meta-embedding. This is a radical and destructive simplification, collapsing multiple types into a single representation – nonetheless, the model is still able to recover a non-trivial percentage of lemmas, retaining about 75% of the original AVEP. Applied together, the ablations above reduce the model to a standard (linear) Transformer. The baseline Transformer *collapses*, utterly and completely failing at capturing the task.

Put together, our experiments assert the robustness and generality of our approach and affirm the positive effect of each of our design decisions. These findings suggest that our model's success stems from a combination of structure-faithful representations, and targeted, deliberate architectural adjustments that honor them.

# 6 Conclusion

**In the Future**  Pending community feedback, there is a number of directions we would be eager to pursue going forward. On the dataset front, we would like to optimize the extraction's performance and coverage beyond the standard library. We further acknowledge that while our structured format is highly elaborate, different levels of abstraction and detail come with their own pros and cons, and are inherently biased towards certain tasks and architectures. That said, the optimal setup would likely be to provide multiple representations and allow the end-user the choice of which one to use; we plan to extend the array of options provided by the extraction. On the integration front, the pipeline implemented so far is unidirectional, in the sense of not providing any hooks for feedback or interaction. Given our strong experimental results, we are currently looking for the best integration angle that can allow at least one full back-and-forth between the language and the machine learning backend. In combination's with Agda's auto, this should enable an end-to-end evaluation in terms of proof completion, but also permit the system's use in real-life scenarios, outside of asynchronous offline settings. Finally, a plethora of improvements and optimizations are possible on the modeling front. Among those, we are primarily interested in seeing our approach applied and evaluated on other languages and especially unifying meta-frameworks such as Dedukti [Assaf et al., 2016] to maximize interoperability.

**In Sum**  We have presented a method to expose structured representations of program-proofs extracted from arbitrary Agda libraries and files. Applying the extraction algorithm on the expansive Agda standard library, we have generated and released a high-quality static resource, intended to enable the offline (pre-)training of ML-based proof guidance tools. To our knowledge, this resource is the first of its kind for Agda. Across proof assistants, the resource is also one of the very few that explicates proof structure at the sub-type level rather than just pretty-printed surface representations. Noting the particularities of dependently typed programs, and pinpointing weak points of prior related endeavors, we argued for the difficulty but also the need for structurally faithful models and representations. Utilizing the ultra-high resolution of our extracted resource, alongside recent advances in efficient and structurally biased self-attention, we set out to design and implement such a model. Trained for premise selection, the model achieves remarkably high precision while relying just on type structure alone, experimentally confirming both the utility of the resource and the soundness of our approach. Contrary to recent trends in ML in general and ML-guided theorem proving in particular, our model *tiny*, requires no pretraining or server hosting, and makes no use of autoregressive language modeling. We open-source our code and make the extracted resources and trained model weights accessible to the community.

## Ethical Impact

This paper presents work aiming to advance the fields of Representation Learning and Interactive Theorem Proving. We do not see any potential risks for misuse or negative social impact.

## Acknowledgments and Disclosure of Funding

This publication is based upon work from COST Action CA20111: European Research Network on Formal Proofs (EuroProofNet), supported by COST (European Cooperation in Science and Technology). Konstantinos gratefully acknowledges Josef Urban for his helpful feedback and encouragement.

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

# A  Dataset Extraction

## A.1  Extraction Example

Let us clarify the data extraction process by means of an illustrative example: translating the Agda program of Figure 1 to the corresponding JSON structure of Figure 4.

The resulting JSON in Figure 4 contains the imports about equality in its `scope-global` field (*i.e.,* both their types and proof terms), the publicly exported definitions of this example under `scope-local`, and the private lemma that allows us to commute `suc` over addition under the `scope-private` fields only.

```
{ "scope-global":
    [ {"name": "Agda.Builtin.Equality._≡_<12>", ...}
    , {"name": "Agda.Builtin.Equality._≡_.refl<20>", ...}
    , ...],
  "scope-local":
    [ {"name": "ℕ<2>", ...}
    , {"name": "ℕ.zero<4>", ...}
    , {"name": "ℕ.suc<6>", ...}
    , {"name": "_+_<8>", ...}
    , {"name": "+-comm<20>"
      ,"type": "(m n : ℕ) → (m + n) ≡ (n + m)"
      ,"definition": ...
      ,"holes":
        [...
        , { "ctx": "(n : ℕ)"
          , "goal": "(zero + suc n) ≡ (suc n + zero)"
          , "term": "cong suc (+-comm zero n)"
          }
        , ...
        ]
      }
    ]
  "scope-private": [{"name": "+-suc<38>", ...}]}
```

(a) Subset of extracted JSON fields.

```
data ℕ : Set where
  zero : ℕ
  suc  : ℕ → ℕ

_+_ : ℕ → ℕ → ℕ
_+_ = ...

+-comm : (m n : ℕ) → m + n ≡ n + m
+-comm zero    zero    = ...
+-comm zero    (suc n) = {!!}
+-comm (suc m) zero    = ...
+-comm (suc m) (suc n) = ...
  where
    +-suc : ∀ m n → m + suc n ≡ suc (m + n)
    +-suc = ...
```

(b) Interactive Agda session focusing on the hole of Figure 4a.

Figure 4: JSON extract of the Agda code of Figure 1.

Zooming further into the holes recorded for the commutativity proof, in particular the second case where $m = 0$ and $n' = n+1$, we can see that the captured *context* contains a bound variable $n$ arising from the pattern of the second argument. By virtue of dependent pattern matching, the *goal* type has now been refined to a more precise one, namely $0 + (1 + n) = (1 + n) + 0$. Finally, we record the term that successfully fills the hole of the aforementioned type cong suc ( +-comm zero $n$ ), from which we can extract just the lemmas used (*i.e.,* cong, suc, and +-comm) specifically for the task of premise selection. Note the numbers attached to each definition's name, which are immediately drawn from Agda's internal name generation process and ensures that references to lemmas are unique.

## A.2  Extraction Statistics

The extraction algorithm is developed against version 2.6.3 of Agda[5] and applied on version 1.7.2 of the Agda standard library[6]– an open-source, community-developed collection of definitions and proofs for general purpose coding and proving with Agda. The library's contents span subjects ranging from abstract algebra, common (as well as exotic) data types, functions, relations, *inter alia*.

In what follows, we select and present a few statistics pertaining to the extracted data that are particularly relevant to our experimental setup.

Figure 5a presents the empirical distribution of file sizes in terms of scope entries (global imports or local definitions) and holes. Global and local scope entries roughly follow a bell-curve with long

---

[5]https://agda.readthedocs.io/en/v2.6.3/
[6]https://github.com/agda/agda-stdlib/releases/tag/v1.7.2

right tails: a handful of files contain more than 1,000 definitions in total. Holes are more evenly distributed, except for a high peak at 0 due to files without any holes.

Figure 5b similarly presents the empirical distribution of the AST lengths of the types of lemmas and (context-merged) holes. Both distributions are bell-shaped, with holes being in general larger. As before, a handful of ASTs are overlong, counting thousands of tokens.

Figure 5c, finally, presents the empirical distribution of lemma occurrences. We obtain this by considering the set of fully qualified lemma names as they occur in the extracted files, and associating each name with two occurrence counts: the number of times the name is encountered across files as an import (*i.e.,* excluding the single time it is defined), and the number of times this lemma is actually selected as a relevant premise by a hole. We only count a single occurrence per hole (*i.e.,* we increment the counter by just 1, even if that lemma is used multiple times). As expected, the plot suggests that most lemmas are only ever used locally, with linearly less lemmas being imported exponentially more times. Analogously, most lemmas rarely ever find their way to a hole, with linearly less lemmas being useful in the derivation of exponentially more holes. This goes to show that premise selection is *hard*: a few lemmas might be relevant "universally", but the relevance of most lemmas is sparse and contextually decided.

### A.3 Limitations

The dataset is by no means set in stone and is amenable to various generalizations. Since Agda is a dependently-typed language, terms appear in all kinds of places, *e.g.,* as type-level indices of an inductive datatype. Currently, only terms appearing in functions (*i.e.,* proofs) are considered, but there is nothing preventing us in principle from utilizing terms that appear elsewhere.

While the extractor's performance is sufficient for our studies so far, the current implementation is in no way the most efficient possible. First, there is a lot of context sharing between adjacent holes that could be leveraged to drastically reduce disk usage. Second, CPU time and RAM consumption quickly exhibit exponential behavior due to multiple sub-invocations of Agda's type checker, making it impossible to extract data from a handful of complex Agda modules. Finally, the dataset does not currently contain structured decompositions for literals, sorts (*i.e.,* kinds of types), and universe levels. We have conciously ignored them for the time being, seeing as they are largely boilerplate for proof search purposes, but acknowledge that they still might prove useful when reasoning about type hierarchies.

## B  Experimental Setup

### B.1 Tokenization

We read and tokenize files offline to facilitate training. Figure 2 depicts the structure of the JSON file of Figure 4 (in turn the extract of the Agda code of Figure 1) as it is perceived post-tokenization by the neural model. Definitions (proof-level terms) are ignored, and type ASTs are binarized. References (presented in gray font) are resolved, and treated as pointers to either other ASTs or nodes within the same AST, depending on whether the referenced object is another lemma or a variable. Type operators and meta-symbols (presented in normal font) are treated as static elements of a fixed vocabulary. The presentation is the product of some artistic license: in reality, sets are quantified by universe levels, and the equivalence operator is a reference to the actual definition of equivalence.

### B.2 Data Filtering & Splitting

We fix an 85%-15% training-evaluation split over the 818 files extracted from the Agda standard library. After tokenization, we reject 18 files (2.2%) containing mutual inductions, *i.e.,* cyclic reference structures which our model does not yet support. Another 196 files (23.97%) contain no holes, and are thus of no use in this present setup. Finally, we identify 36 files (4.4%) representing size anomalies, each accumulating more than $2^{14}$ tokens. According to our original split, 31 of those were marked for training and 5 for evaluation. We ignore the former, but use the latter as a standalone left-out test, intended to stress the model's generalization. Finally, we inspect the fully qualified names of scope entries and find that the training sets contains 10,320 unique definitions. The evaluation set contains 3,495, with 732 of them being "novel" in the sense of never occurring in the training set (the rest being imported from the training set, or, conversely, having been imported at least once by the training set). All holes in the evaluation set are by construction novel, seeing as they stem from locally defined scope entries.

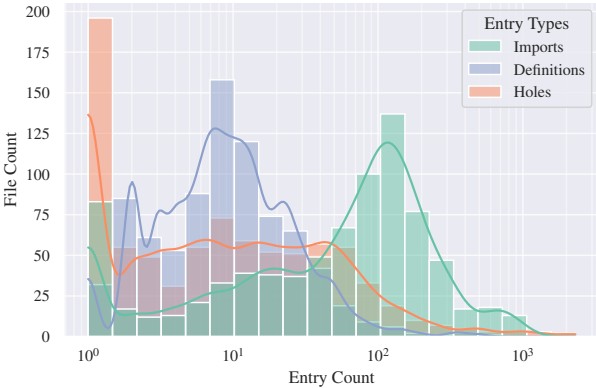

(a) Histograms and KDE plots of the counts of imports, definitions and holes across files, offset by 1 and log-transformed.

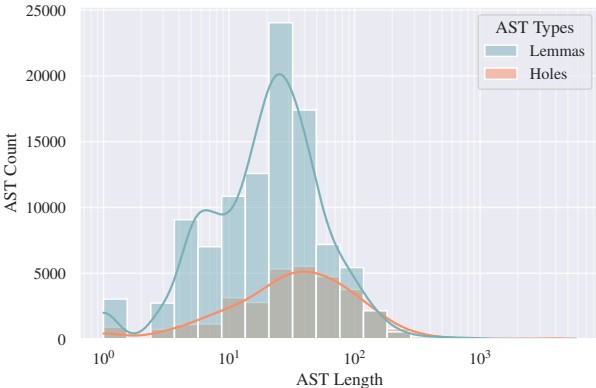

(b) Histograms and KDE plots for the AST lengths of the types of lemmas and holes across files, log-transformed.

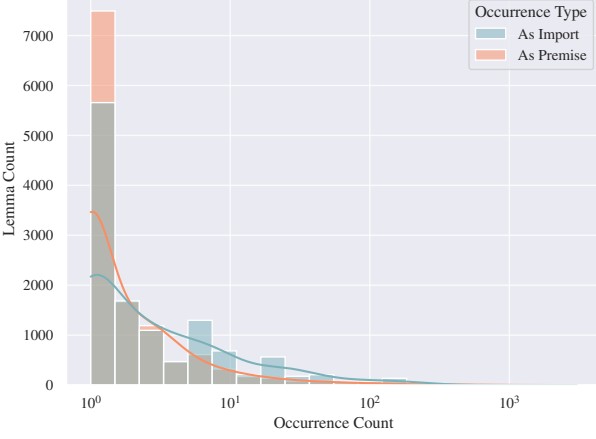

(c) Histograms and KDE plots for the occurrence counts of each unique lemma, offset by 1 and log-transformed.

Figure 5: Statistics of the extracted data relevant to our experimental setup.

| PARAMETER | VALUE | PARAMETER | VALUE |
|---|---|---|---|
| model dim | 256 | algorithm | AdamW($\beta_1 := 0.9, \beta_2 := 0.99, \lambda := 1e^{-2}$) |
| # layers | 6 | schedule | linear warmup (3) – cosine decay (97) |
| # heads | 8 | lr curve | $0 \ldots 5e^{-4} \ldots 1e^{-8}$ |
| head dim (queries, keys) | 16 | batch size (files) | 1 |
| head dim (values) | 32 | batch size (holes) | 32 |
| feed-forward dim | 1024 | dropout | reference labels, post-attention, post-ffn (0.1) |

| (a) Model hyperparameters. | (b) Optimization hyperparameters. |
|---|---|

Table 2: Hyperparameter setup

## B.3 Training

We generate training batches by randomly selecting a fixed number of files. From each file, we extract the entire scope along with a predetermined number of holes. We use the encoder to obtain vector representations for the types of both the holes and the lemmas within scope. Next, we consider all pairs in the Cartesian product of lemmas and holes within each file but discard causally illegal pairs– those with a hole referencing a lemma that has not yet been defined. For each pair, the gold data provide us with a label indicating whether the indexed lemma actually occurs when filling in the indexed hole. We use this label as an imperfect proxy for the lemma's relevance, *i.e.,* whether it *should* be used to fill in the indexed hole. Given a hole, we want QUILL to be able to identify *all* relevant lemmas; that is, to successfully discriminate between positive and negative pairs. Translating this requirement to an objective function, we make a minor modification to infoNCE [Oord et al., 2018] so as to capture the existence of possibly multiple positive matches. Concretely, denoting with $P_h^+$ and $P_h^-$ the sets of positive and negative pairs anchored to a common hole $h$, we compute a per-hole loss term $\mathcal{L}_h$ as:

$$\mathcal{L}_h := \sum_{p \in P_h^+} -\log \frac{\exp\left(f(p)\right)}{\exp(f(p)) + \sum_{n \in P_h^-} \exp(f(n))} \tag{3}$$

where $f$ is simply a weighted dot-product reducing each vector pair to a scalar value. We derive a per-batch loss by aggregating the loss terms of all the holes considered.

We conduct 5 training iterations using the same model and optimization hyperparameters, reported in Table 2. We conduct no hyperparameter sweep of any kind, instead opting for efficiency with a sensibly sized model and otherwise default optimization values that maximize GPU utilization. We train on a single A100 for 100 epochs, monitoring training and evaluation performance. Keeping at-home deployment in mind, we opt for a modestly sized model. QUILL enumerates fewer than 6 million parameters in total– more than 3 orders of magnitude smaller than the LLM-based architectures currently in use. Once trained and binarized, it takes up a mere 26MB of disk space. Training completes in about 16 hours, but converges halfway through by epoch $55\pm7$, overfitting slowly thereafter.

## C  JSON format

The dataset has a non-trivial JSON structure, so we deem it worthy to document it more precisely here: Fig. 6 presents the JSON *schema* in a more readable BNF-grammar form. We slightly simplify things by embedding definitions, types, patterns, and term expressions, all in a single syntactic category of "terms" (TERM). While this is not completely accurate (*e.g.,* it does not make sense to have a datatype declaration appear as a function's clause pattern), it strikes a good balance between accuracy and accessibility; the JSON *validation* that accompanies the JSON schema will make sure that such conditions are always met.

A JSON *file* (FILE) contains scope entries and holes for each sub-term appearing in the source program, where the scope is split into three categories: the *global scope* of imported definitions; the *local scope* of definitions declared earlier in the same file; and the *private scope* of helper definitions that is only visible to the current definition (*e.g.,* arising from a where clause). Each *scope entry* (SC_ITEM) has its type and the actual definition that inhabits this type. *Holes* (HOLE) are only included for the local or private scope entries, since the modules imported in the global scope already

$\langle$FILE$\rangle$ ::= {*name:* **string**
    *scope-global:* $\langle$SC_ITEM$\rangle^*$
    *scope-local:* $(\langle$SC_ITEM$\rangle,\langle$HOLE$\rangle)^*$
    *scope-private:* $(\langle$SC_ITEM$\rangle,\langle$HOLE$\rangle)^*$ }
$\langle$SC_ITEM$\rangle$ ::= {*name:* **string**
    *type:* $\langle$TERM$^+\rangle$
    *definition:* $\langle$P_TERM$\rangle$ }
$\langle$HOLE$\rangle$ ::= {*context:* $\langle$CTX_ITEM$\rangle^*$
    *goal:* $\langle$TERM$^+\rangle$
    *term:* $\langle$TERM$^+\rangle$
    *premises:* $\langle$string$\rangle^*$ }
$\langle$CTX_ITEM$\rangle$ ::= {*name:* **string**
    *pretty:* **string**
    *type:* $\langle$TERM$^+\rangle$ }
$\langle$TERM$^+\rangle$ ::= {*original:* $\langle$P_TERM$\rangle$
    *simplified:* $\langle$P_TERM$\rangle^?$
    *reduced:* $\langle$P_TERM$\rangle^?$
    *normalised:* $\langle$P_TERM$\rangle^?$ }
$\langle$P_TERM$\rangle$ ::= {*pretty:* **string**
    *term:* $\langle$TERM$\rangle$ }
$\langle$TERM$\rangle$ ::= $\langle$REF_DB$\rangle$ | $\langle$REF_SC$\rangle$ |
    $\langle$PI$\rangle$ | $\langle$LAM$\rangle$ | $\langle$APP$\rangle$ |
    $\langle$ADT$\rangle$ | $\langle$CON$\rangle$ | $\langle$REC$\rangle$ | $\langle$FUN$\rangle$ |
    $\langle$LIT$\rangle$ | $\langle$SORT$\rangle$ | $\langle$LEVEL$\rangle$
$\langle$CLAUSE$\rangle$ ::= {*ctx:* $\langle$TERM$\rangle^*$
    *patterns:* $\langle$TERM$\rangle^*$
    *body:* $\langle$TERM$\rangle$ }

$\langle$REF_DB$\rangle$ ::= {*tag:* "DeBruijn"
    *index:* **number** }
$\langle$REF_SC$\rangle$ ::= {*tag:* "ScopeReference"
    *name:* **string** }
$\langle$PI$\rangle$ ::= {*tag:* "Pi"
    *name:* **string**
    *domain:* $\langle$TERM$\rangle$
    *codomain:* $\langle$TERM$\rangle$ }
$\langle$LAM$\rangle$ ::= {*tag:* "Lambda"
    *abstraction:* **string**
    *body:* $\langle$TERM$\rangle$ }
$\langle$APP$\rangle$ ::= {*tag:* "Application"
    *head:* $\langle$TERM$\rangle$
    *arguments:* $\langle$TERM$\rangle^*$ }
$\langle$ADT$\rangle$ ::= {*tag:* "ADT"
    *variants:* $\langle$TERM$\rangle^*$ }
$\langle$CON$\rangle$ ::= {*tag:* "Constructor"
    *reference:* **string**
    *variant:* **number** }
$\langle$REC$\rangle$ ::= {*tag:* "Record"
    *context:* $\langle$TERM$\rangle^*$
    *fields:* $\langle$TERM$\rangle^*$ }
$\langle$FUN$\rangle$ ::= {*tag:* "Function"
    *clauses:* $\langle$CLAUSE$\rangle^*$ }
$\langle$LIT$\rangle$ ::= {*tag:* ... **string**}
$\langle$SORT$\rangle$ ::= {*tag:* ... **string**}
$\langle$LEVEL$\rangle$ ::= {*tag:* ... **string**}

Figure 6: JSON schema of datasets extracted via AGDA2TRAIN.

have a corresponding JSON dataset with their corresponding holes. For each hole, we record the *context* of (typed) bound variables, its type, and the term that successfully fills the hole (from which we also compute the set of used lemmas, specifically for the task of premise selection).

Definitions can either declare an *algebraic datatype* (ADT) consisting of various types of constructors; a *record* (REC) with fields of certain type; or a *function* (FUN) that can be defined via pattern matching across multiple *clauses* (CLAUSE).

The rest of the cases correspond to the usual presentation of terms in $\lambda$-calculus: variable references (REF_DB/REF_SC); abstractions (PI/LAM); applications (APP); and primitives for literals/sorts/levels (LIT/SORT/LEVEL). We do not retain the structure of literals, sorts, and levels; there is no real use for it on premise selection, but it can easily be added in the future if the need arises.

Last, notice how top-level terms are printed to human-readable strings (P_TERM), or even captured at different points of their evaluation (TERM$^+$). Each of these reduction levels are options (marked with $^?$), since evaluating an arbitrary term might prove too costly; only those computed within a reasonable time frame are populated.

