# OpenReview forum: "Learning Structure-Aware Representations of Dependent Types"
_NeurIPS.cc/2024/Conference — NeurIPS 2024 poster_

### Official Review · Reviewer_xpMx · 2024-07-09

**Soundness:** 3
**Presentation:** 3
**Contribution:** 3
**Rating:** 7
**Confidence:** 4

**Summary:**

The article contributes to the field by detailing a method to capture and utilize the internal compilation states of Agda, a dependently typed programming language, through JSON files. These JSON files represent different compilation states, reflecting various stages of the coding process. Key contributions of the article include:

- It explains how each JSON file mirrors Agda's internal state during compilation, which is dynamic and changes based on the current focus of the compilation process. This approach captures the evolution of the file content at different coding stages.

- The method involves iterating through each definition within a file, focusing on all possible sub-terms rather than just potential prefixes of a proof. This detailed navigation allows for a more comprehensive utilization of the file as a data resource, enhancing the understanding of Agda’s compilation process.

- By treating each file as a data resource and navigating its states preemptively, the approach acts like a type-safe data augmentation routine. This maximizes the resource utilization of the file content.

- The article also highlights how each state's typing context is recorded, including the type, relevant definitions, and local variables. Additionally, it thoroughly explicates the structural aspects of terms in Agda, going beyond simple λ-expressions to include a variety of constructs like datatype declarations and pattern matching, which are all captured within the JSON structure.

- For better readability and to aid practitioners interested in a more linear format, the dataset includes pretty-printed displays of each abstract syntax tree (AST).

The article provides foundational insights into leveraging Agda’s internal mechanisms for better data handling and representation in JSON format, which could be useful for both researchers and developers working with dependently typed languages.

**Strengths:**

- This article provides a detailed investigation of the role of representation learning in Automated Theorem Proving (ATP), which is novel.
- It provides valuable resources for the community.

**Weaknesses:**

- Although it's within the setting of premise selection, I still hope to provide a usable baseline for search-based ATP.
- Can you provide some visual results on representation?

**Questions:**

- Can you provide some visual results on representation?
- Are you asking if it's possible to commit to open-sourcing the code in the future?

**Limitations:**

- Premise selection is an interesting area, but it cannot truly generate theorems. It could be considered for expansion into a more general setting.

---

> ### Author Rebuttal · Authors · 2024-08-06
>
> Hi, and thank you for taking the time to review our paper.
>
> > `... visual results on representation?`
>
> We provide a visualization of the empirical distributions of positive and negative lemmas in Figure 3, which indicates that positive lemmas are consistently ranked significantly higher than negative ones in the development set. We further supply additional visualizations in the author rebuttal pdf. We apologize if this format is inconvenient; we have produced interactive plots that are much more informative, but we have no way to make them available (author guidelines restrict us from uploading url links, even if anonymized; we have asked the area chair for permission, but we have received no response yet).
>
>
> > `... open-sourcing the code in the future?`
>
> We have made our code available for review as supplementary material. We are committed to keeping our code open source and freely available after the reviewing process is concluded.
>
> > `... search-based ATP ... more general setting`
>
> Indeed these are future directions we also wish to pursue, but definitely lie outside the scope of this paper.
> Although it should have been clear from the introduction that premise selection is just the most natural first step, we will also extend the conclusion of Section 6 to explicitly mention these points for future work.

---

> > ### Comment · Reviewer_xpMx · 2024-08-09
> >
> > **Thank you for your response.**
> > - I would like to know the details of how you construct multiple positive samples.
> > - What does "negative pairs anchored to a common hole h" mean? Are the negative samples selected using the in-batch trick?
> > - I would like to know if, as long as a suitable method for constructing positive and negative samples is used, the architecture becomes less important for the premise selection task, and a standard bidirectional encoder model can also be effective. I believe that a vanilla Transformer combined with a contrastive learning objective is also an important baseline.

---

> ### Author Response · Authors · 2024-08-09
>
> Thank you for your questions, we really appreciate your engagement.
>
> * `multiple positive samples`
>
> We don't have to do anything special! Think of some make-do proof that looks like `A (B C)`, where `A`, `B` and `C` are lemmas. There are several possible holes: `? (B C)`, `A (? C)`, `A (B ?)` and `A ?`. The last one contains *two* positive lemmas: `B` and `C`, so we want both of them to be picked by the premise selector.
>
> *  `... negative samples selected ...`
>
> Consider the previous example, and a context providing availability to lemmas `A`, `B`, `C`, `D` and `E`. For this present hole, `A`, `D` and `E` are *negative* as they do not appear inside of it.
>
> * `... vanilla Transformer ...`
>
> We were also wondering the same thing; we already used a vanilla Transformer to catastrophic results. See the last row of Table 1. (ps: see also line 354)

---

> > ### Comment · Reviewer_xpMx · 2024-08-09
> >
> > - **negative samples selected:** I see. So, actually, the negative samples are randomly selected excluding the positive samples?
> > - **Supplement the baseline:** Is the vanilla Transformer in the last row of Table 1 trained with autoregressive loss? I don't see any experimental results comparing the vanilla Transformer with Equation 3 in Table 1.

---

> > > ### Author Response · Authors · 2024-08-09
> > >
> > > * No, we pick **all** negatives (in the current file). We experimented briefly with picking only hard/easy/random negatives, but this vanilla setting proved to work quite well.
> > > * No, it's trained using the exact same experimental setup, *i.e.*, premise-selection head and the modified infoNCE loss.

---

> ### Comment · Reviewer_xpMx · 2024-08-09
>
> - **Supplement the baseline:** I see. Can you provide results from training a pre-trained model such as BERT in conjunction with Equation 3? Although the comparison may not be entirely fair, scaling the model from the this paper is also challenging, so the best approach is to combine it with an existing representation model. In the future, is it possible to scale up models designed with your architecture?
> - **Ablation study:** I suspect the model's performance comes from the position encoding. Could you provide an ablation study using a vanilla Transformer combined with binary-tree positional encoding[1] and Equation 3?
>
> **Reference:**
> [1]: Kogkalidis, K., Bernardy, J. P., & Garg, V. (2023). Algebraic Positional Encodings. arXiv preprint arXiv:2312.16045.

---

> ### Author Response · Authors · 2024-08-09
>
> * It is not clear how to connect BERT with the current dataset/experimental setup, can you elaborate? We remind you that the experimental setup does not involve textual representations of any kind.
> * We have **already** provided this ablation; please inspect Table 1 (row: `- structured attention`), and paragraph `Ablations` in Section 5.2. As you say, a large part of the model gains are indeed due to the structured attention from the tree-structured positional encodings.

---

> ### Comment · Reviewer_xpMx · 2024-08-09
>
> **Supplement the baseline:**: Considering the remarkable effect of this positional encoding[1], I would like to see whether it is possible to inject this structural bias by taking a pre-trained model like BERT and continuing the training using contrastive learning with this positional encoding. If possible, could you provide the experimental results of vanilla transformers combined with other positional encodings, such as Yarn[2] or RoPE[3]?
>
> **Reference:**
> [1]: Kogkalidis, K., Bernardy, J. P., & Garg, V. (2023). Algebraic Positional Encodings. arXiv preprint arXiv:2312.16045.
> [2]: YaRN: Efficient Context Window Extension of Large Language Models.
> [3]: Su, J., Ahmed, M., Lu, Y., Pan, S., Bo, W., & Liu, Y. (2024). Roformer: Enhanced transformer with rotary position embedding. Neurocomputing, 568, 127063.

---

> > ### Author Response · Authors · 2024-08-09
> >
> > We remark once more that our model is not directly compatible with or comparable to text-based LLMs (autoregressive, like GPT, or encoder-only, like BERT). The model does *not* operate on textual token sequences, but on complex structures made of a (really) small number of primitive symbols (namely: `Π`, `->`, `λ`, `@`, `Sort`, `Level`, `Literal`, `deBruijn`), plus references.
> >
> > That said, we can still attempt to train a YaRN-based sequential version, but from-scratch (rather than pretrained).

---

> ### Author Response · Authors · 2024-08-09
>
> > `In the future, is it possible to scale up models designed with your architecture? `
>
> We missed this one, sorry. The most honest answer is "it depends". There are projects ongoing, like Dedukti (see Conclusion), which attempt a common formalization interface/meta-language between different type-theory-based proof assistants. If/when such a project takes hold, and upon larger libraries becoming accessible, it would make sense to upscale the model. Until then, and for the time being, it is very likely that there's little benefits from model scaling given the current data availability. Otherwise, from an architectural perspective, there's nothing really stopping one from training larger (or smaller) variations of the proposed model.

---

> ### Comment · Reviewer_xpMx · 2024-08-09
>
> **Scaling the model**: Given the slow growth of the Agda library and limitations of Dedukti , why not consider choosing Mathlib instead? It has a vast amount of data and mature methods for acquiring large datasets, such as [1], which make scaling easier.
>
> **Reference:**
> [1] Han, J. M., Rute, J., Wu, Y., Ayers, E. W., & Polu, S. (2021). Proof artifact co-training for theorem proving with language models. arXiv preprint arXiv:2102.06203.

---

> ### Author Response · Authors · 2024-08-09
>
> That's a fair question -- the datasets used in the cited paper are all without exception **text-based**. We kindly ask you to inspect pages 17 - 20 of the linked paper, which present example datapoints. The data points presented are pretty-printed proofs (*i.e.*, strings), making them suitable for autoregressive language modeling. The data points we work with are *not* "strings that look like proofs" -- they are the "structural footprints of proofs". We explicitly refer to this this paper (in line 106), and compare their dataset extraction and modeling methodologies to ours (line 117+, and Section 4.2).

---

> ### Author Response · Authors · 2024-08-10
>
> Hi again. We return with a **RoPE-based baseline**, as requested.
>
> | metric | stdlib[id] | stdlib[ood] | unimath | type-topo |
> | -------- | ----------- | -------------- | ---------- | ------------- |
> | AP | 22.56 | 12.83 | 18.00 |9.13 |
> | R@P | 11.74 | 8.95 | 8.33 |3.82 |
> | R@1 | 11.78 | 16.16 | 9.75 |3.22 |
>
> As you suggested, performance is **much better** than the vanilla sinusoidal positional encoding scheme. At the same time, it is still **much worse** than the tree-structured positional encoding we use (or any ablation, for that matter). Neither of the two observations is too surprising:
> * we know RoPE empirically outperforms sinusoidal encodings from the literature
> * RoPE is still a sequential encoding scheme; types are not sequences, and sequential models can't do them justice (section 4.2)

---

> > ### Comment · Reviewer_xpMx · 2024-08-10
> >
> > Thank you for your experiments.  I believe there are other ways to inject structural bias into an autoregressive model through RoPE, such as using a Mask matrix to block certain connections between tokens.

---

> > > ### Author Response · Authors · 2024-08-10
> > >
> > > Thanks once more for your engagement, we appreciate the ongoing discussion.
> > >
> > > You could certainly use a boolean mask to block out specific attention coefficients (*e.g.* you could attend to just "descendents" in the AST, or even something more elaborate), but it's not immediately clear whether this could faithfully capture the AST structure. Going for something more radical (*e.g.* attending to just some local neighborhood of nodes, given an appropriate definition of local) would result in the problems we attribute to GNNs (lines 211 - 217). Conceptually, this *should* also prove inadequate for capturing the long distance dependencies commonly found in "real life" dependent types (even though empirically this *could* also turn out not to be the case).
> > >
> > > Did you have something specific in mind? Is there some experiment that you'd like us to do along these lines (keeping the time window in mind, of course)

---

> > > > ### Comment · Reviewer_xpMx · 2024-08-13
> > > >
> > > > Thank you for your response. I think these models[1][2] can serve as baselines. Could their experiments be supplemented?
> > > >
> > > > **Reference:**
> > > > [1]Wang, Y. S., Lee, H. Y., & Chen, Y. N. (2019). Tree transformer: Integrating tree structures into self-attention. arXiv preprint arXiv:1909.06639.
> > > > [2]Nguyen, X. P., Joty, S., Hoi, S. C., & Socher, R. (2020). Tree-structured attention with hierarchical accumulation. arXiv preprint arXiv:2002.08046.

---

> ### Author Response · Authors · 2024-08-13
>
> We address the limitations of hierarchical tree methods conceptually in lines 205-211. Practically, the papers you suggest are not suitable baselines due to their reliance on quadratic $O(n^2)$ attention, which is infeasible for our setup as discussed in lines 221-228. With lemma lengths ranging from $10^0$ to $10^4$ tokens and files counting $10^0$ to $10^3$ lemmas, quadratic attention results in $O(mn^2)$ complexity (for a batch size of 1), making it untenable. This is exactly why we chose a linear attention kernel (lines 233 - 248).
>
> While hierarchical tree and linear attention could be combined theoretically, no existing implementation that we know of exists. We sincerely apologize, because your suggested experiments are valid and interesting, but we cannot implement, train and evaluate such a mechanism within the 12 hours left before the discussion period closes. Please also keep in mind that training alone takes approximately 6-8 hours (assuming the compute resources are immediately available).

---

### Official Review · Reviewer_2Qh1 · 2024-07-10

**Soundness:** 3
**Presentation:** 1
**Contribution:** 3
**Rating:** 7
**Confidence:** 2

**Summary:**

This paper is the first to extract data from Agda, a dependently typed programming language. The extracted data can be utilized by machine learning practitioners. Additionally, the paper proposes a novel neural structure designed to faithfully represent dependently typed programs. The experiments demonstrate the effectiveness of the proposed neural architecture.

**Strengths:**

- The dataset extracted in this paper is the first based on the formal language Agda, which will significantly benefit the neural theorem proving community by providing more formal data.
- The proposed neural representation structure performs very well compared to vanilla transformers, demonstrating strong ablation results. It offers a better inductive bias and completeness.

**Weaknesses:**

I don’t see any major weaknesses in this paper, except that it is somewhat difficult to follow. It would definitely benefit from more illustrative figures to explain various concepts. Concepts in Agda, such as dynamic compilation, “holes,” typing contexts, and term structures, are hard to understand. Including figures with concrete examples of these components, in addition to textual descriptions, would be very helpful.

**Questions:**

- Typos in line 162 `that that`.

**Limitations:**

The authors have adequately addressed the limitations and posed no negative societal impact.

---

> ### Author Rebuttal · Authors · 2024-08-06
>
> Hi, and thanks for the review.
>
> We’re very glad you appreciated our work. We acknowledge that the presentation can sometimes be dense. We aimed to keep the tone and language inclusive and to provide brief descriptions of any jargon where possible. However, the paper's topic sits at a narrow intersection of type theory, automated theorem proving, and ML, which unfortunately requires some familiarity with all three.
>
> Although we hoped the concrete examples and figures in the Appendix would cover such concerns, it turns out that wasn't the case.
> To this end, we commit to extending section A.2 with some more explanations revolving around the topics you mention in your comment.

---

### Official Review · Reviewer_ctFo · 2024-07-10

**Soundness:** 2
**Presentation:** 2
**Contribution:** 3
**Rating:** 6
**Confidence:** 3

**Summary:**

The paper reported the creation of a new dataset to predict a term for filling the hole of a proof in Agda. The paper also proposed an attention-based neural architecture, trained it on the made dataset, and shows it outperforms the ordinary Transformer.

**Strengths:**

+ Creating a new dataset for the problem of proof term predication on Agda.
+ The paper proposed an original attention-based model for proof term prediction and showed that it beaves more well than Transformer.

**Weaknesses:**

- The importance of creating a new dataset could be discussed more appropriately and carefully. Specifically, I'd like to see why the dataset is necessary or valuable although there exist the datasets for automated theorem proving already (like the ones mentioned in the related work section and MiniF2F).  I don't think it is sufficient just to say "there doesn't exist a dataset for Agda" because then one just could adopt the non-Agda datasets.
- The experimental setting is unclear for me. Specifically, how is Transformer set up? Is it fine-tuned on the Agda dataset? If so, it is trained against the data in JSON format or text format? If the JSON format is adopted, I suspect it might be unfavorable for Transformer because it needs to parse JSON data by itself.
- I'm unsure Transformer is appropriate as a baseline because, even if the models are restricted to be Transformer-based, there exist many previous works that claim their models outperform the vanilla Transformer as, e.g., the following [1]

[1] Stanislas Polu, Ilya Sutskever: Generative Language Modeling for Automated Theorem Proving. CoRR abs/2009.03393 (2020).

- The presentation could be improved. The following are unclear for me.
  - What is the outputs of the proposed model? An AST of a proof term? Or, just a text?
  - What is the design principles of the proposed model? Why did the authors think efficient and structured attention work well on the task of interest?
  - In Figure 3, what the x- and y-axis represent, respectively?
  - What is "the added cognitive load" mentioned in Section 5.2 (page 9)?

**Questions:**

* Why is creating a dataset on Agda important?
* Why did the paper choose the vanilla Transformer as a baseline?
* Was the Transformer fine-tuned? If so, how?

**Limitations:**

Certain limitations are described in the paper.

---

> ### Author Rebuttal · Authors · 2024-08-05
>
> Hello, and thank you for your critical review. We will try to address some of your concerns below.
>
> ---
>
> > ` ...why the dataset is necessary or valuable ...`
>
> Thank you very much for this question. The dataset is radically different to existing datasets, and while do point the fact out (*e.g.*, in lines 8, 58 - 59, 152 - 157, 161 - 167, 376, 381, 384 - 385 *etc*.), it is true that we should have been more explicit about its uniqueness. It is unfortunate that we couldn't convey the points through our paper; let us try to do so here instead.
>
> The dataset is **necessary** because:
> * It is the *first dataset* that we know of that explicates the shape of program-proofs at the *subtype* level in a way that is *structurally preserving*. Whereas most datasets contain *textual representations* of formal proofs, our dataset exposes also the *real underlying structure* of the proof objects under scrutiny. As we extensively argue in Section 4.2 (lines 185 - 205), it is exactly this structure that distinguishes a *proof* from a *string that happens to look like a proof*. Not having access to this structure and a dataset that explicates it means the community is confined to working with the wrong kind of models and evaluating on the text-only benchmarks.
>
> The dataset is **valuable** because:
> * it represents proof objects according to Agda's term/type syntax, which is mind boggingly close to the real syntax of the underlying type theory. The type theory is a universal theoretical foundation; Agda is an implementation. Having representations close to the theoretical foundation means that the dataset has more chances of finding use in other related endeavors than a dataset that capitalizes on the surface syntax of *e.g.* Lean proofs.
> * Along with Coq, Agda is one of the two *most important programming languages* for *foundational mathematics* on the planet today. Homotopy type theory and the univalent foundations program, i.e. the contemporary approaches to foundational mathematics, rely predominantly on these two languages to formalize and verify their findings. Expediting the progress of these two languages and their interface to ML is *expediting the progress of mathematics*.
> * There is *epistemic value in scientific plurality*. By providing a dataset that highlights the structural elements of proofs in Agda, we enable the exploration of *new* and *diverse* approaches to proof verification and automated reasoning.
>
> > ` ... one just could adopt the non-Agda datasets ... `
>
> We would just like to point out that this is largely non-trivial. There is no methodology that we know of (or can imagine) that would allow the conversion of text-formatted proofs of Coq into structure-faithful representations of Agda programs. The contrary is somewhat more likely.
>
> > `The experimental setting is unclear for me.`
>
> We are very sorry to hear. Would that perhaps warrant a reduction in your *very high confidence* score? In either case, we will try to answer your questions below.
>
> > `Is it fine-tuned on the Agda dataset?`
>
> The transformer is **trained** on the Agda dataset; we start from a blank slate (no pretraining) (line 298).
>
> > ` ... is trained against the data in JSON format or text format?`
>
> Neither. The model takes `a collection of ASTs as input` (Section 4.2, line 301, *etc*.). The input to the model is the *literal* structure of the underlying Agda program. Each AST is a collection of *position-specified type operators* (zeroary, unary, binary) or *references* (intra- or inter-AST).
>
>
> > `If the JSON format is adopted, I suspect it might be unfavorable for Transformer because it needs to parse JSON data by itself.`
>
> Very good point! This is **exactly** our point as well, except we are arguing against **all text-based representations** (Section 4.2). What we feed the model is *neither* the JSON format, *nor* the pretty printed strings of proof objects: it is their *actual* structure.
>
> > `I'm unsure Transformer is appropriate as a baseline because ... `
>
> Also a very good point, and one we ourselves make in Section 4.2 (lines 185 - 188). The vanilla Transformer, and all LLM-related endeavors suffer from exactly these biases, regardless of whether they are generative or encoder-only. This is why we think our work is an important alternative take: one that emphasizes structural discipline and *tools fit for the task* rather than *tasks fit for the tool*.
>
> > `What is the outputs of the proposed model? An AST of a proof term? Or, just a text?`
>
> Neither; our model ` produces a scalar value for each hole-lemma pair ` (line 301). Without the premise selection head attached to its top, the model produces a neural representation for each AST (hole or lemma).
>
> > `What is the design principles of the proposed model?`
>
> The design principles are extensively motivated in Section 4.2. Basically: sequential architectures are not fit for the task; people simply use them because this is what seems to work now. Tree architectures are nice, but they are expensive and hard to parallelize and perform badly. Graph architectures are not a good conceptual match for the task either; the input is not just any arbitrary graph, but a very refined and structurally consistent one. Therefore we need an architecture that does the problem's structure justice. Our architecture does exactly that, and, from what it looks like, it works.
>
> > `Figure 3`
>
> This is a paired **violin plot**. It shows the empirical distribution of the scores of positive ground truth items (above) vs the scores of negative ground truth items (below). It shows that positive items have a consistently higher score than negative items. We explain this in lines 318 - 324.
>
> > `What is "the added cognitive load" mentioned in Section 5.2 (page 9)?`
>
> This is exactly referring to your earlier point; the model now has to *parse* sequentialized type representations, because it misses the structural positional encodings that make this parse structure explicit.
>
> ---
>
> We hope this is helpful.

---

> ### Comment · Reviewer_ctFo · 2024-08-09
>
> Thanks for the response. It resolves some of my concerns, especially the ones for the contributions on the dataset, so now I'm leaning towards acceptance.
>
> As a critical reviewer, I recommend the revision clarifies the contributions described in the response for the dataset, in the introduction. I couldn't find these contributions in the introduction of the submission, which prevents me from understanding the benefits of the new dataset compared with the existing ones (especially reading the related work section, which mentions the internal type structure as a difference, but what it means was unclear). The response says that, with the quote of lines 58 - 59, the contributions are mentioned in the introduction. Perhaps the crucial word in the lines is "program-proofs," but it is difficult, at least for me, to find that this phrase means that the dataset made by the paper is more structured (and in what sense) than the existing datasets.
>
> Furthermore, even after admitting the differences from the existing dataset, it is still not clear to me what the "subtype level" means. Does it mean what is called "inter-AST" in Section 4.2? In any case, it would be helpful to clarify what it means and clearly connect the contributions described in the introduction with the details in the remaining sections.
>
>
> >> I'm unsure Transformer is appropriate as a baseline because ...
>
> > Also a very good point, and one we ourselves make in Section 4.2 (lines 185 - 188). The vanilla Transformer, and all LLM-related endeavors suffer from exactly these biases, regardless of whether they are generative or encoder-only. This is why we think our work is an important alternative take: one that emphasizes structural discipline and tools fit for the task rather than tasks fit for the tool.
>
> I don't think this response addresses my concern of why only the vanilla LLM is used as a baseline and the other LLM-related models (claimed to better perform than the vanilla LLM) is not used.

---

> ### Author Response · Authors · 2024-08-09
>
> Thank you for your engagement, and for revising your rejection.
>
> > ` ... "subtype level" .. `
>
> Yes, that's correct! This is indeed one of the benefits of the extra structure. For example, consider the dependent type $\Pi_{x:A}(B~x)$, or in Agda syntax, `(x : A) -> B x`, which denotes a family of types `B` indexed by objects of type `A`. The dataset represents this as an AST that looks like:
>
> ```
> DependentProduct
> |--- Argument
> |    |--- Type: A  (this would actually be an inter-AST pointer to the AST defining A)
> |--- Body
>      |--- FunctionApplication
>           |--- FunctionBody
>                |--- Type: B  (this would be an inter-AST pointer to the AST defining B)
>           |--- FunctionArgument
>                |--- @0  (this would be an intra-AST pointer to the Argument above)
> ```
>
> This illustrates how structural relations are fully specified at the subtype level, all the way down to typing primitives. This is what allows the neural system to operate on a structural rather than textual level. Practically, our embedding layer only has the tiniest number of symbols (*i.e.*, only primitive type operators). Neural processing happens over complex structures made of these symbols.
>
>
> > `..LLM is used as a baseline and the other LLM-related models`
>
> We do not use a LLM. We use a (non-pretrained) tiny, customized Transformer made of 6 million parameters. This is **1000 times less** parameters than the LLMs that you cite and are currently in use (lines 563 - 566). The trained model is 26MB big. To make a fair architectural comparison, we compare against a Transformer of a similar size, because **this** is the architecture these LLMs rely on. Our comparison demonstrates that our model dramatically outperforms similarly sized Transformers. Comparing with overparameterized, closed-source, pretrained LLMs is beyond the scope of our work and offers no epistemic insights that we can think of.
>
> ---

---

> > ### Comment · Reviewer_ctFo · 2024-08-09
> >
> > Thanks for the clarification for the baseline. I'm not very convinced whether using a customized Transformer as a baseline is appropriate, but it would be helpful to describe in the main text the detail of the baseline and why it is used unless I missed an explanation in the main text of the submission.

---

> > > ### Author Response · Authors · 2024-08-09
> > >
> > > You're very welcome!
> > >
> > >  The model we revert to after all ablations in Table 1 is no longer customized -- it is basically a run-of-the-mill vanilla Transformer. It is a useful baseline exactly because:
> > > 1. it is the most common architecture empowering modern state of the art LLMs that find use in ATP tasks, and
> > > 2. it is what our model boils down to when we remove all of the task-adapted components

---

### Official Review · Reviewer_XfNT · 2024-07-11

**Soundness:** 3
**Presentation:** 3
**Contribution:** 4
**Rating:** 7
**Confidence:** 4

**Summary:**

This paper introduces a learning algorithm for Agda, a functional language used  in proof assistance that is known for its dependent types.
The paper includes the design of the algorithm, and an experimental evaluation,

**Strengths:**

The Agda language has become very popular in the functional prog community, especially because of dependent types.  This results in very structured programs. and I believe in a real challenge for ML. The authors design a relatively simple system to address this problem, but claim to achieve very interesting results. I believe that given the popularity of Agda and the results achieved, this is a solid contribution.

The authors have a very clear writing style,

**Weaknesses:**

The major weakness of the paper is that it assumes familiarity with Agda, The authors drop Figure 1 in the paper but make no effort to explain it. They assume the reader to know what are dependent types. One or two paragraphs would make a difference here.

This makes it hard to understand the results (especially for non-experts like me). You define a positive to be an hole that is filled by a lemma in the proof? Does this mean that a hole/lemma pair could have been a positive, it just wasn´t considered? Also, is this for any proof? If this is a proof assistant, shouldn´t it be about a specific hole in a specific sequence of steps? , In B.1 you do mention scope and selecting a fixed number of holes. can you be more specific? Also how many positive and negative examples do you actually have

It would also be interesting if we could have a baseline. say. are there comparable resuls for, say, Coq?

Finally in "Practical' you seem to consider either Tree Based Models or Transformers. Any specific reason to ignore GNNs?

**Questions:**

I mentioned above my main suggestions. Other details:

Fig 1, please explain

1.1 universal: ie language that uses type theory as its foundational core. It would be nice to mention a few such languages?

Page 6: it seems that the Taylor expansion is how you achieve global features, which seems to be goal here. Yet, later on you drop Taylot expansion and still obtain good results.

J. Rute, M. Olsˇak, L. Blaauwbroek, F. I. S. Massolo, J. Piepenbrock, and V. Pestun. Graph2tac: ´
442 Learning hierarchical representations of math concepts in theorem proving, 2024.
443 N -> Missing venue

**Limitations:**

The authors do not see potential negative societal impact, and I would agree.

---

> ### Author Rebuttal · Authors · 2024-08-05
>
> Hello, and many thanks for your honest review.
>
> ---
>
> We're glad you appreciated the value of the architecture and the potential in the results.
>
> We acknowledge the weaknesses you spot. We tried our best to accommodate a reader that is not necessarily familiar with Agda, but does have prior exposition to ATP concepts and type theory. Given the focused nature of our contribution, it proved really hard to find a more inclusive tone/vocabulary without making the presentation entirely superficial.
>
> > `... a positive to be an hole that is filled by a lemma ... a hole/lemma pair could have been a positive, it just wasn´t considered?`
>
> This is precisely on point. A positive is the specific lemma that fills that hole. Another lemma could possibly have filled that hole, but in practice the vast majority of "other" lemmas in context would not have been appropriate.
>
> > ` ... shouldn´t it be about a specific hole in a specific sequence of steps?`
>
> That's a good question, and we're very happy you asked. As we explain in lines 73 - 74, the cool thing about Agda is that you don't have to write your proofs/programs sequentially -- you can write the entire proof as normal, but defer some parts to the future: these are holes. That means that a hole is not like an element of a *sequence* with left-only context, but an element of a syntax *tree* with a much wider context, spanning (i) the rest of the tree, but also (ii) all other trees in scope. This is precisely the factor motivating the design of our dataset *and* the architecture, and what makes both stand out from the established literature:
> 1. On the dataset front, we consider *all* proofs of the Agda libraries we train on, and we turn *all* possible subproofs within each proof into a hole.
> 2. On the architecture front, we take care to properly encode the entire context in a structure-aware fashion.
>
> > `... fixed number of holes ... more specific`
>
> Our training configuration considers a batch of 32 holes-per-file, 1-file-per-batch. This is not the result of hyper-parameter optimization; it's just a value that maximized resource utilization without causing cuda OOM errors.
>
> > ` ... how many positive and negative examples do you actually have`
>
> Across the training dataset, we have 53,668 positives and 15,026,433 negatives, *i.e.*, positive lemmas make for 0.35% of the total lemmas.
>
> > `Any specific reason to ignore GNNs?`
>
> We do in fact have a (very) short discussion on GNNs in lines 211 - 217. The idea is that GNNs capture the structure implicitly and in small, localized neighborhoods by virtue of the graph convolution kernel. If the graph is large (which *is* the case when dealing with complex dependent types), the model doesn't have a reliable way to account for the whole type's structure, which leads to coarse and structurally unrefined representations.
>
>
> > `Fig 1, please explain`
>
> We are defining the type of naturals, N, and equip it with a binary additive operation. There are two ways to make a natural; either the $zero$, for which $zero + n = n$, and, inductively, via the $suc$ (succedent) function, for which it holds that $(m + 1) + n = (m + n) + 1$. We then proceed to prove that given $m$ and $n$ naturals, $m + n$ is equivalent to $n + m$. The proof follows a case-by-case analysis:
> 1. If $m$ and $n$ are both $zero$, we can prove our goal using reflexivity (both sides of the equation are *syntactically* equal).
> 2. If $m$ is $zero$ and $n$ is actually the succedent of some other $n'$, then we can invoke the proof of the commutativity of $m + n'$.
> 3. Similar to (2): If $n$ is $zero$ and $m$ is the successor of some other $m'$, then we can invoke the proof of the commutativity of $m' + n$.
> 4. If $m$ is the successor of some other $m'$ and $n$ is actually the successor of some other $n'$, then we appeal to a helper lemma about how the successor function $suc$ distributes across addition, before finally invoking the proof of the commutativity of $m + n'$.
>
> > `It would also be interesting if we could have a baseline. say. are there comparable resuls for, say, Coq?`
>
> This would be very difficult to accomplish, given the vast difference between the datasets and the inherent information available in the actual content within those datasets.
>
> Having said that, our methodology is, at least conceptually, applicable to any language based on Type Theory, hence it is possible to port our constructions to Coq and then start considering comparisons against previous works.
>
>
> > `1.1 universal: ie language that uses type theory as its foundational core. It would be nice to mention a few such languages?`
>
> Duly noted, we will refer to the most commonplace functional programming languages and theorem provers based on Type Theory, e.g. Haskell/OCaml/Coq/Agda/Lean.
>
>
> > `... Taylor expansion ...`
>
> The effect of the Taylor expansion step is by no means small, offering an *absolute* increase of 3.2 points in precision and 4.1 in recall. The fact that this increase is overshadowed by the relative gains offered by structured attention and the structural variable representation serves only to empirically verify the necessity of the architectural desiderata we argued for in Section 4.2, and to show that their effect is significantly stronger than that of SOTA architectural adjustments and micro-optimizations.
>
>
> ` ... Missing venue ... `
>
> Good catch, thank you! We had to cook up the citation manually, seeing as Graph2Tac was released concurrently with our submission.
>
> ---
>
> We hope this clarifies things a bit. Once again, thank you very much for your time and effort.

---

> > ### Comment · Reviewer_XfNT · 2024-08-13
> >
> > Thanks for the nice replies

---

> > > ### Author Response · Authors · 2024-08-13
> > >
> > > Thank you for the nice questions :)

---

### Author Rebuttal · Authors · 2024-08-07

We thank all reviewers for their time and effort. We have responded to each review individually.

Here, we attach a pdf containing visualizations of neural representations, as requested by reviewer xpMx. Pending AC approval, we would be happy to include an anonymized link to an interactive version of these visualizations.

Figures 1 to 9 display dimensionality-reduced (TSNE) representations of the subset of the Agda stdlib we trained on. Lemmas are colored according to the (sub-)library they belong to. We show slices along major axis pairs (xz, yz, yx) with three filtered data views for each slice: the full dataset (all nodes and references), libraries #2 to #5 (ranked by size), and libraries #6 to #9. Library #1 is omitted due to its large size and mostly uniform distribution, which renders the figure illegible. The visualizations reveal strong clustering according to different libraries. Note that the neural representation algorithm does not distinguish lemmas based on names or (names of) references; this clustering is purely emergent. Figure 10 presents a heatmap of similarity scores between two random lemma sets. Pairwise similarity is generally low, with three notable exceptions: one pair defines upper semilattices across different objects, another refers to symmetric function inversion properties, and a third pair coincidentally consists of the same object twice.

---

### Decision · Program_Chairs · 2024-09-25

**Decision:**

Accept (poster)

**Comment:**

There has recently been a great deal of interest in using neural networks (and LLMs in particular) for theorem proving.  There are several formal languages in common use, including Coq, Agda, and Lean, all of which are based on the calculus of constructions, which is a lambda calculus with dependent types.

Current SOTA when applying language models to code of any kind is to model only the surface syntax of terms.  Despite the fact that the underlying structure of source code is an abstract syntax tree, LMs typically ignore tree structure.  Moreover, LMs pay undue attention to syntactic features like the spelling of variable names, even though formal theories always treat terms as equivalent under alpha-renaming -- which means that it is the structure of the term, not the spelling of variable names, that matters.  As a result, there is a severe mismatch between the way in which LMs are usually applied, and the underlying formal theory used for theorem proving.

This paper takes some initial steps in correcting this mismatch.  The authors extract the complete internal representation of Agda terms into a JSON format.  This includes not only the tree-structure of terms, but also typing information, goals, lexical scope, etc.  The authors release the processed JSON as a new dataset.  The authors then design a custom transformer which operates on this structure, using tree positional encodings.  They show that the structured representation improves over a vanilla transformer on the task of premise selection.

The reviewers universally recommend "accept" for this paper, and I concur.   The new dataset alone will be useful to other practitioners in this area.  In my opinion, however, the most important contribution of this paper is not necessarily the dataset, or even the details of the architecture and experiments.  The main contribution of this paper is to explain and discuss why the structure of terms is so important in formal languages.  This is an area which up until now has been almost completely ignored, and which will guide future research in LM-based theorem proving.

The main weakness of the paper, as with many papers that use new datasets and architectures, is the size of the model that the authors are able to train.